# Colistin: Lights and Shadows of an Older Antibiotic

**DOI:** 10.3390/molecules29132969

**Published:** 2024-06-21

**Authors:** Erica Diani, Gabriele Bianco, Milo Gatti, Davide Gibellini, Paolo Gaibani

**Affiliations:** 1Department of Diagnostic and Public Health, Microbiology Section, University of Verona, Strada Le Grazie 8, 37134 Verona, Italy; erica.diani@univr.it (E.D.); davide.gibellini@univr.it (D.G.); 2Department of Experimental Medicine, University of Salento, 73100 Lecce, Italy; gabrielebnc87@gmail.com; 3Department of Medical and Surgical Sciences, Alma Mater Studiorum, University of Bologna, 40126 Bologna, Italy; milo.gatti2@unibo.it

**Keywords:** colistin, lipopeptide, antimicrobials, antimicrobial resistance

## Abstract

The emergence of antimicrobial resistance represents a serious threat to public health and for infections due to multidrug-resistant (MDR) microorganisms, representing one of the most important causes of death worldwide. The renewal of old antimicrobials, such as colistin, has been proposed as a valuable therapeutic alternative to the emergence of the MDR microorganisms. Although colistin is well known to present several adverse toxic effects, its usage in clinical practice has been reconsidered due to its broad spectrum of activity against Gram-negative (GN) bacteria and its important role of “last resort” agent against MDR-GN. Despite the revolutionary perspective of treatment with this old antimicrobial molecule, many questions remain open regarding the emergence of novel phenotypic traits of resistance and the optimal usage of the colistin in clinical practice. In last years, several forward steps have been made in the understanding of the resistance determinants, clinical usage, and pharmacological dosage of this molecule; however, different points regarding the role of colistin in clinical practice and the optimal pharmacokinetic/pharmacodynamic targets are not yet well defined. In this review, we summarize the mode of action, the emerging resistance determinants, and its optimal administration in the treatment of infections that are difficult to treat due to MDR Gram-negative bacteria.

## 1. Introduction

Antibiotic resistance represents a serious public health threat and is associated with millions of deaths annually [1]. Since the discovery of the first antimicrobial molecules, the emergence of novel traits of resistance to antimicrobials has been observed concomitantly [2]. It is well known that antimicrobial resistance is associated with the misuse and overuse of related drugs in different fields of applications (humans, animals and plants). Indeed, the presence of antimicrobial rich environments creates favorable conditions that allow the selection of resistant subpopulations in opposition to sensitive microorganisms [3].

With the diffusion of these drugs and the rapid increase in antimicrobial resistance, the development of microorganisms resistant to multiple antimicrobial classes of compounds has been subsequently observed [4]. The emergence of multidrug-resistant (MDR) microorganisms imposes different limitations on clinicians by reducing the available antimicrobial armamentarium. In the last years, the diffusion of MDR strains, especially Gram-negative bacteria, has been considered an urgent threat that requires a prompt response. To overcome these limitations, several strategies have been adopted, including new schemes of treatment combining antimicrobial molecules with no activity alone and the development of novel antimicrobial molecules [5,6]. At the same time, the revival of older antibiotics considered as last resort drugs has posed new prospective issues in the treatment of difficult-to-treat (DTR) infections due to MDR strains [5]. 

Colistin, also known as polymyxin E, is an old antimicrobial molecule that was discovered in the middle of the 19th century in Japan from a culture of the *Paenibacillus polymyxa* subspecies [7]. Colistin is a cyclic oligopeptide antimicrobial belonging to the class of polycationic antibiotics and it is active against most Gram-negative bacteria by binding to the lipopolysaccharide (LPS) of the outer cell membranes by electrostatic interaction. The linkage between colistin and the outer membrane created a disorganization of the outer membrane’s structure thus results in an alteration of the outer membrane and consequent intracellular content release and bacterial death [7].

In the last years, the renewal of older antibiotics such as colistin has created a new perspective in the treatment of DTR infections [6,8]. However, the emergence of new traits of resistance to this drug and its adverse toxic effects to mammalian cells has mitigated its use in clinical practice [6,7].

In this review, we discuss the principle of the mode of action, the emerging traits related to the resistance, and the use of colistin in clinical practice from pharmacological and clinical point of views. The main points discussed in this review are shown in Table 1.

## 2. Mechanisms of Action, Antibacterial Activity and Adverse Effects

### 2.1. Structure and Mode of Action

Colistin is an amphiphilic lipopeptide antibiotic, discovered in 1947 by Koyama [9,10]; it is produced by *Paenibacillus polymyxa subspecies colistinus.* Colistin, also called polymixyn E, is a member of the polymyxin family of antibiotics. In 1952, the first formulation in clinical use was a solution for intravenous administration that showed its bactericidal function against many Gram-negative bacteria, but not against Gram-positive, anaerobic bacteria or mycoplasmas. Due to its potent antibacterial activity against Gram-negative bacteria, colistin was initially considered a “miraculous molecule”. However, since the 1970s, its use in clinical practice has been mitigated due to its severe adverse effects [11]. The original molecule has been principally modified to reduce the nephrotoxicity effect, and two forms of colistin are clinically available for human treatment: colistin sulfate and colistin methanesulfonate, also called colistimethate sodium (prodrug form). Differences between these compounds are related to their use and toxicity. In particular, colistin sulfate is an active compound that is administered topically and orally, while colistimethate is used in formulations administered by parenteral and nebulization routes. 

Colistin’s basic structure consists of a core region formed by a hydrophobic portion, which is a cyclic heptapeptide linked by a tripeptide bridge to the fatty acids. The colistin molecule is positively charged due to the presence of five diaminobutyric acid residues linked to the core [12]. The prodrug form differs from this structure due to the presence of methan-sulfonates linked to the diaminobutyric acids (Figure 1) [13].

It is noteworthy that the five diaminobutyric acid residues, which confer the positive charge to the molecule, play a determining role in the drug’s antibacterial effect, which is generally described with the Shai–Matsuzaki–Huang (SMH) model [14,15,16] and represented in Figure 2.

Colistin acts by competition with and the displacement of Ca^2+^ and Mg^2+^ from the negatively charged sulfate portion of the lipid A in the lipopolysaccharide molecule (LPS) of Gram-negative bacteria (Figure 2a). This ionic dislocation by colistin seems necessary for forming pore-like structures [17,18,19]. The loss of binding ions and their substitution with colistin molecules alters the tertiary structure of LPS, creating the possibility for colistin itself to insert its own portion of fatty acids into the membrane, definitively compromising the permeability of the outer membrane (Figure 2b). In addition, colistin acyl fat, inserted into the bilayer, alters the inner membrane stability, leading to bacterial membrane disruption and a bactericidal effect [20]. Colistin also plays a key role in preventing endotoxin-induced shock by binding to the lipide A portion [21]. This drug acts both at the surface and intracellular levels, in particular altering the vesicle–vesicle contact of bacterial cells. In brief, colistin crosses the membrane and causes the fusion of the inner leaflet of the outer membrane and the outer leaflet of the cytoplasmic membrane, disrupting the cytoplasmic bilayer, altering the osmotic balance, and leading to cell death [22,23] (Figure 2c). The antibacterial action of colistin was also reported at the molecular level, where it can induce oxidative stress and, consequently, DNA, protein, and lipid damage in bacteria through ROS production, being able to inhibit essential enzymes involved in the respiratory chain, such as the NADH-quinone oxidoreductase [24], leading to cell death.

### 2.2. Adverse Effects

Colistin treatment was dismissed in clinical use, principally due to its nephrotoxicity effect, which is lower for the prodrug form (i.e., colistin sodium methanesulfonate). Its adverse effects were principally due to its re-absorption by proximal tubule cells through an endocytotic process, mediated by megalin, and through facilitative transport by two transporters located in the apical cell membrane, the human peptide transporter 2 (PEPT2), and the carnitine/organic cation transporter 2 (OCTN2) [25,26]. The intracellular accumulation of colistin induces mitochondrial and endoplasmic reticulum stress, with consequent toxic cellular effects [27]. This mechanism leads to cellular lysis and acute tubular necrosis [28,29]. The incidence of colistin-induced acute kidney injury varies between 12.7 and 70% in intensive care unit patients [30,31,32,33]. A recent study by Kilic and colleagues demonstrated that the nephrotoxicity effect depends proportionally on the duration of treatment and is prevalent in older patients [28,34]. 

Due to the high lipid content of neuronal cells, colistin could also exert its action in these cells, and some patients (with an incidence of about 7% [34]) experienced neurological adverse effects, such as paresthesia, seizures, confusion, ataxia, and visual disturbances [35]. The mechanism by which colistin induces these effects is a non-competitive presynaptic myoneuronal blockade of acetylcholine release [36]. Its adverse effects on neuronal cells could be reverted by discontinuing the therapy.

### 2.3. In Vitro Antimicrobial Activity

The in vitro activity of colistin was tested with success on *Acinetobacter baumannii*, a large part of *Enterobacteriales,* and *Pseudomonas aeruginosa* [37]. In particular, for 106 non-duplicate isolates of *A. baumannii*, the authors reported a minimum inhibiting concentration of 0.5 μg/mL for MIC_50_ and of 1.0 μg/mL for MIC_90_ in monotherapy [37].

Walkty et al. [38] analyzed the colistin antibacterial activity exerted on 3480 isolates of Gram-negative bacilli from patients recruited over 2 years in 12 hospitals in Canada (CANWARD Study). In this study, the authors reported a MIC_90_ value ≤2 μg/mL against several clinically relevant Gram-negative bacilli, such as *Escherichia coli* (1732 isolates), *Klebsiella* spp. (515 isolates), *Enterobacter* spp., *A. baumannii*, and *P. aeruginosa* (561 isolates), including all 76 MDR *P. aeruginosa* isolates tested in the CANWARD Study. 

A cross-sectional and descriptive study conducted on 52 MDR *P. aeruginosa* isolates, collected from urine, pus specimens and respiratory tract, reported a MIC_50_ value of 1.0 μg/mL and a MIC_90_ value of 3.0 μg/mL [39].

In the preceding years, several studies reported on the activity of colistin in combination other antimicrobial molecules. In particular, colistin, in combination with meropenem or tigecycline, showed synergistic activity against colistin-resistant KPC-producing *K. pneumoniae* [40]. In a study based on an in vitro checkerboard assay, Kheshti and coworkers [41] reported the good synergistic activity of colistin treatment in combination with ciprofloxacin, levofloxacin (5%—the lowest level), imipenem, meropenem, and ampicillin–sulbactam molecules and its higher synergism in combination with rifampin (55%) when tested on 20 isolates of *A. baumanii*. 

A recent study [42] conducted on 219 *K. pneumoniae* isolates demonstrated the strong synergistic effects of minocycline and colistin on colistin-resistant and minocycline-intermediate or -resistant *K. pneumoniae*. This drug combination acts by disrupting the outer membrane (by colistin) without affecting the cytoplasmic membrane, allowing the entrance and accumulation of minocycline at the intracellular level.

### 2.4. Antimicrobial Susceptibility Testing

The chemical structure of colistin and its cationic charge make the use of classical susceptibility tests, like E-tests and disc diffusion, difficult. To overcome this limitation and to provide a pharmacological alternative to the numerous multi-resistant bacterial species, classical diagnostic protocols have been modified, allowing the measurement of colistin susceptibility. Broth microdilution, the gold standard for colistin susceptibility test, is modified using a cation-adjusted Muller–Hinton broth without adding a surfactant [43,44]. Another method that has been approved by the CLSI for only *Enterobactarales* and *Pseudomonas* spp. is a broth disc elution, modified by Simner and colleagues [45]. The test, renamed as Colistin Broth Disc Elution (CBDE), is easier than the Broth microdilution and is based on analysis of the efficacy of a graded concentration of colistin (of 1,2,4 μg/mL) obtained from colistin disc elution in 10 mL of cation-adjusted Muller–Hinton broth, tested on a 0.5 Mc Farland of bacteria. EUCAST colistin breakpoints table, version 14.0, reports the following cut-off value for the detection of phenotypic resistance: an MIC of 2 mg/L for *Enterobacterales* and for *Acinetobacter* spp., and an MIC of 4 mg/L for *P. aeruginosa*.

## 3. Mechanisms of Colistin Resistance

A variety of mechanisms may be involved in the acquisition of colistin resistance in Gram-negative bacteria, and they can be summarized into four groups (Figure 3): the modification of LPS structures by chromosomal mutations (i), modification of LPS structures by acquisition of plasmids (ii), the loss of LPS structures (iii), and the overexpression of efflux pumps (iv). 

### 3.1. Modification of LPS Structure by Chromosomal Mutations

A reduction in the negative charge of lipid A of LPS leads to the loss of electrostatic interactions with colistin and consequently to resistance [46]. Many genes and operons are involved in LPS modifications: (1) *pmrC* and *pmrE* genes and the *pmrHFIJKLM* operon, which promote the addition of phosphoethanolamine (PEtn) and/or 4-amino-4-deoxy-L-arabinose (L-Ara4N) to lipidA; (2) regulatory two-component systems such as PmrAB, PhoPQ, and crab; and (3) *mgrB*-negative regulator genes.

The addition of L-Ara4N and/or PEtn to lipidA changes the negative charge of the cell membrane by neutralizing the negatively charged phospholipids [47,48,49,50]. In detail, the addition of PEtn to the 1′- or 4′-phosphate group of lipid A is carried out by PmrC, a putative membrane protein with phosphoethanolamine transferase activity encoded by pmrABC operons [11,51,52,53,54]. The synthesis of L-Ara4N from uridine diphosphate glucuronic and its addition to lipid A is promoted by *pmrHIJKLM* operons (also called *arnBCADTEF*) and PmrE activity [51]. PmrB is a cytoplasmic membrane-bound protein which activates PmrA by phosphorylation, and PmrA in turns activates the regulation of the pmrABC and *pmrHFIJKLM* operons and the *pmrE* gene. Subsequently, these operons and genes lead to LPS modification by adding PEtn and L-Ara4N to lipid A [55]. Although the L-Ara4N modification of LPS has been described as a common mechanism of colistin resistance among Gram-negative bacteria (*Klebsiella pneumoniae*, *Escherichia coli*, *Salmonella enterica*, and *Pseudomonas aeruginosa*), it does not occur in *Acinetobacter baumannii* because it lacks all the genes required for L-Ara4N biosynthesis [51]. Alternatively, the addition of galactosamine to the 1′-phosphate position of lipid A, following activation of the sensor kinase PmrB, is associated with moderate levels of colistin resistance in *A. baumannii* [52].

The mutation of PmrA/PmrB results in the upregulation of the *pmrABC* and *pmrFHIJKLM* operons and *pmrE* gene. This leads to PEtN modification of lipid A, and in turn, results in colistin resistance. Several mutations have been reported in many Gram-negative bacteria, such as *Salmonella enterica* [56,57], *K. pneumoniae* [58,59,60], *A. baumannii* [61,62,63], *P. aeruginosa* [64,65], and *E. coli* [57,66,67].

The transcription of *pmrFHIJKLM* operons is also activated by the PhoPQ regulatory two-component system. PhoQ is a sensor kinase that promotes the expression of the regulator protein PhoQ, which promotes *pmrFHIJKLM* operon transcription via phosphorylation. Furthermore, PhoP indirectly activates pmrA through the PmrD connector protein, which subsequently activates the transcription of the *pmrHFIJKLM* operon. This then leads to the synthesis and transfer of PEtn to lipid A [11,49,50]. The mutation of the *phoP/Q* genes. which that leads to acquired colistin resistance, has been identified in *K. pneumoniae* and *E. coli* [68,69,70]. Higher polymyxin MICs have been observed in PhoQ-deficient *P. aeruginosa* mutants when additional alterations affected other regulatory two-component systems (CprRS and ColRS) [71].

More evidence has accumulated on the role played by *mgrB*, a gene encoding a small regulatory transmembrane protein that exerts negative feedback on the kinase activity of PhoQ [72]. The inactivation of *mgrB* leads to the activation of a phosphorylation cascade involving chain PhoQ, PhoP, PmrD and/or PmrAB. This finally triggers the expression of the *pmrHIJKLM* operon, resulting in LPS modification. The mutation of *mgrB*, including point mutations, deletions, nonsense, and insertion sequences (IS*5*-like, IS*1F*, IS*Kpn14*, IS*Kpn13*, IS*10R*), represents the most common mechanism of colistin resistance in clinical *K. pneumoniae* isolates [69,73,74,75,76,77]. The wide range of resistance levels showed by Gram-negative strains harbouring mutations in the genes *pmrAB*, phoPQ, or *mgrB* suggests a role for other genetic loci. Mutations in the CrrAB two-component system has been associated with increased levels of colistin resistance in strains of *K. pneumoniae* [74,78]. The mutation/inactivation of the *crrB* gene led to the activation of the *pmrHFIJKLM* operon and the *pmrC* and *pmrE* genes through the overexpression of the *pmrAB* operon [77,78]. Furthermore, various PEtn-coding genes, such as *eptA* (*pmrC*), *eptB* (*pagC*), and *eptC* (*cptA*), are able to add PEtn to LPS and can be involved in colistin resistance [79]. The overexpression of eptA has been associated with colistin resistance in *A. baumannii* [61,80]. Gerson et al. showed that mutations in the *eptA* gene (R127L and ISAba1 insertion) were associated with the overexpression of EptA and colistin resistance in *A. baumannii* [61]. 

### 3.2. Loss of LPS Structure

The complete loss of lipid A or the LPS core, leading to colistin resistance, has been observed in *A. baumannii*. The analysis of laboratory-induced colistin-resistant *A. baumannii* showed that the high level of resistance to colistin was caused by the inactivation of LPS biosynthesis genes *lpxA*, *lpxC*, *lpxD*, and *lpsB* [81]. Various nucleotide substitutions, deletions, and insertions that cause frameshifts or result in truncated proteins have been reported for in vitro mutants and clinical isolates [81,82,83,84]. Moreover, the disruption of *lpxC* and *lpxD* via the insertion of IS elements was described in colistin-resistant *A. baumannii* isolates [81,82,83,84,85,86]. Although LPS loss is an effective mechanism of colistin resistance, it has significant fitness costs, and this explains why these mutants are rarely encountered in the clinical setting [87].

### 3.3. Plasmid-Mediated Colistin Resistance

Since the first report of the *mcr* gene encoding for phosphoethanolamine transferase (*mcr-1*) in *E. coli* in China in 2015 [88], several reports worldwide have demonstrated the presence of *mcr-1* and additional 9 families (*mcr-2* to *mcr-10*), with more than 100 overall variants of different Gram-negative species distributed worldwide [11,49,50,51,52,53,89,90,91,92,93,94]. MCR is a member of the PETN enzyme family, located mainly in the bacteria plasmids, and its activity results in the modification of lipid A via PETN addition. The enzyme has a domain inserted into the inner membrane and a periplasmic C-terminal sulfatase catalytic domain. 

In 2018, Partridge et al. proposed a nomenclature for *mcr* genes. Several variants were identified, especially for MCR-3 and MCR-1 [95]. MCR-1 and MCR-2 share 81% identity at the amino acid sequence level. Sequence identity suggests that these two variants originated from *Moraxella* spp. [89], with *mcr-3*, *mcr-4,* and *mcr-7* coming from *Aeromonas* spp. and *Shewanella frigidimarina*, respectively [90,91,92,93].

The *mcr*-1 variant can be connected to various types of plasmids, including IncHI2, IncI2, IncX4, IncP, IncX, and IncFIP. The *mcr-2* gene was detected on an IncX4 plasmid. The presence of insertion sequences (IS*Apl1*, IS*1595)* on the genetic environment of *mcr* genes explains the possibility of their integration into bacterial chromosomes [94].

Plasmid-mediated colistin resistance represents the mechanism of greatest concern because of the ease of intra- and inter-species spread. Despite most of MCR-harboring microorganisms belonging to the *Enterobacterales* order, such as *E. coli*, *Salmonella* spp., and *K. pneumoniae*, several reports showed the presence of *mcr*-genes in non-fermenting Gram-negative species such as *P. aeruginosa* and *A. baumannii* complexes [96,97,98,99,100,101,102,103,104,105,106,107,108,109,110]. The *mcr-1* gene is the most commonly detected in *P. aeruginosa* in both clinical [96,97,98,99] and animal settings [100,101,102,103], followed by *mcr-5* [104,105]. In *A. baumannii complex*, the *mcr-1* and *mcr-4.3* are the major variants observed in clinical isolates from Asia and Europe [98,99,106,107,108,109]. Other *mcr* genes found in *A. baumannii* include *mcr-2* and *mcr-3* [110].

### 3.4. Overexpression of Efflux Pumps

The role of efflux pumps in colistin resistance is suggested by a few studies. Efflux pumps, such as the KpnEF and AcrAB, have been reported in *Enterobactericeae*. The ΔKpnEF mutants showed increased susceptibility to various cationic antimicrobial peptides such as colistin [111]. On the other hand, AcrAB is a part of the AcrAB–TolC complex and its overexpression has been observed in colistin-resistant *E. coli*, *K. pneumoniae,* and *Salmonella* strains [112,113,114].

The contribution of EmrAB efflux systems to colistin resistance in *A. baumannii* was shown by in vitro experiments with the ΔemrB mutant [115]. Moreover, the upregulation of the gene-encoding protein components of efflux pumps (*adeI*, *adeC*, *emrB*, *mexB*, and *macAB*) was also observed in colistin-resistant *A. baumannii* strains [83].

The overexpression of the efflux pumps MexXY (RND family) under exposure to ribosome-targeting antibiotics was found to correlate with the increased levels of colistin resistance in *P. aeruginosa* [116]. However, the heterogeneity of MexXY expression observed in clinical isolates of *P. aeruginosa*, showing variable levels of colistin resistance, suggested that the contribution of the efflux pumps to colistin resistance might also be related to other specific genetic backgrounds [117].

Further evidence of the role of efflux pumps in colistin resistance is the suppression of resistance by efflux pump inhibitor (EPI) and cyanide-3-chlorophenylhydrazone (CCCP) in *A. baumannii*, *P. aeruginosa*, *K. pneumoniae*, and *S. maltophilia* [118]. However, a possible explanation is that CCCP-mediated depolarization of the electrochemical gradient may restore the negative charge of the outer membrane and lead to increased susceptibility to colistin [48,118]. Furthermore, various studies suggested a complex regulatory relationship between the efflux pumps and their transcriptional regulators and LPS synthesis, transport, and modification [48].

## 4. Pharmacokinetic/Pharmacodynamic Features

According to several pieces of preclinical evidence, the free area under the concentration-to-time curve to minimum inhibitory concentration ratio (*f*AUC/MIC) was defined as the best pharmacokinetic/pharmacodynamic (PK/PD) target for achieving colistin efficacy in infections caused by *P. aeruginosa* and *A. baumannii* [119]. In a neutropenic murine thigh and lung infection model aimed against three *P. aeruginosa* strains, Dudhani et al. [120] found that the *f*AUC/MIC ratio was the best PK/PD index, correlating with colistin efficacy in both thigh (*R*^2^ = 0.87) and lung infection models (*R*^2^ = 0.89). The colistin *f*AUC/MIC targets required to achieve 1-log and 2-log kills against the three strains were 15.6 to 22.8 and 27.6 to 36.1, respectively, in the thigh infection model, whereas a *f*AUC/MIC ratio ranging from 12.2 to 16.7 and from 36.9 to 45.9 in the lung infection model was found to achieve 1-log and 2-log kills [120]. In a neutropenic murine thigh and lung infection model aimed against three *A. baumannii* strains (of which two were colistin heteroresistant), Dudhani et al. [121] reported that the *f*AUC/MIC ratio was the best PK/PD index, correlating with colistin efficacy in both the thigh (*R*^2^ = 0.90) and lung infection model (*R*^2^ = 0.80). The colistin *f*AUC/MIC targets required to achieve stasis and 1-log kill against the three strains were 1.89–7.41 and 6.98–13.6 in the thigh infection model, respectively, and 1.57–6.52 and 8.18–42.1, respectively, in the lung infection model [121]. Notably, these colistin PK/PD targets, acting against *P. aeruginosa* and *A. baumannii*, were consistent with those retrieved in a recent murine thigh and lung infection model [122]. Indeed, the *f*AUC/MIC ratio was confirmed as the best PK/PD target for predicting colistin efficacy, with *f*AUC/MIC ratios of 7.4–13.7 and 7.4–17.6 required for achieving 2-log kills against *Pseudomonas aeruginosa* and *A. baumannii* strains of, respectively [122]. It should be noted that these PK/PD targets could only be attained in two *P. aeruginosa* strains and in one *A. baumannii* strain in the lung infection model, even at the highest colistin dose tolerated [122].

For *Enterobacterales*, an in vitro model investigated the best PK/PD target of colistin efficacy against three *K. pneumoniae* strains exhibiting MIC values of 0.5, 1, and 4 mg/L, respectively [123]. The *f*AUC/MIC ratio emerged as the best PK/PD target for colistin efficacy, with an *f*AUC/MIC ≥25 being more predictive for a bactericidal effect [123]. Notably, this PK/PD target may be attained at standard colistin doses of 9 MU in 100%, 5–70%, and 0% of *K. pneumoniae* isolates, showing MIC values of 0.5, 1, and 2 mg/L, respectively [123]. These findings may suggest on the one hand the need to revise the current colistin clinical breakpoint against *Enterobacterales*, and on the other hand the potential relevance of implementing a therapeutic-drug-monitoring (TDM)-guided approach for personalizing colistin dosage.

It should be noted that evidence investigating the relationship between optimal PK/PD target attainment for colistin retrieved in preclinical studies and clinical outcomes is currently limited. A prospective observational study investigated the relationship between PK/PD target attainment of colistin and microbiological/clinical outcomes in nine patients affected by multidrug-resistant (MDR) Gram-negative infections (eight caused by *A. baumannii* and one by *K. pneumoniae*) [124]. After the fifth colistin dose of 2 MU, the AUC_0–8_/MIC ranged from 35.5 to 126_._ Although no significant relationship between AUC/MIC ratio and microbiological/clinical cure was found, a positive trend was observed at logistic regression (*p* = 0.28) [124]. A prospective observational study, including 33 patients affected by urinary tract infections and/or pyelonephritis caused by extremely drug-resistant *P. aeruginosa,* reported no significant difference in the *f*AUC/MIC ratio between cases exhibiting favourable clinical outcomes and those with clinical failure (21.5 vs. 47.4; *p* = 0.85) or in the proportion of attainment of an AUC/MIC ratio ≥60 mg/L (32.3% vs. 50.0%; *p* = 0.99) [125]. Undergoing multivariate analysis, the average steady-state colistin concentration showed a trend towards statistical significance for acute kidney injury occurrence (OR 4.36; 95%CI 0.86–20.0; *p* = 0.07—Sorlí et al., 2019) [125].

Studies assessing colistin penetration into different sites of infection are reported in Table 2. Currently, data are only available for the lungs, central nervous system (CNS), and eye (Table 1). Specifically, a prospective observational study investigating the epithelial lining fluid (ELF) penetration of intravenous colistin, administered at a dosage of 2 MU every 8 h in 13 critically ill patients affected by ventilator-associated pneumonia, reported undetectable colistin concentrations in ELF [126].

A prospective observational study including five critically ill patients assessed colistin penetration into cerebrospinal fluid (CSF) administered intravenously at a dosage of 2–3 MU every 8 h [127]. The colistin CSF-to-plasma ratio was 0.05, with the absolute concentrations retrieved in CSF allowing the attainment of optimal PK/PD targets, but only against *P. aeruginosa* and *A. baumannii* strains, showing an MIC value up to 0.06 mg/L [127]. In regard to ocular penetration, currently, only a preclinical animal model has assessed this issue in twenty rabbits receiving intravenous colistin at a dosage of 5 mg/kg [128]. Overall, absolute colistin concentrations were extremely low in.

Aqueous humor and undetectable in vitreous humor in most of the included cases [128].

Overall, these findings strongly support the implementation of alternative agents in the case of deep-seated infections, especially given the limited colistin penetration rate in lung and CSF and the failure in attaining optimal PK/PD targets. Notably, these findings may be expected according to the physicochemical and PK features of colistin, namely its hydrophilic properties, large molecular weight, and limited volume of distribution [129].

## 5. Future Prospectives

Several strategies have been proposed for contrasting the emergence and diffusion of antibiotic-resistance. Since the discovery of the first antimicrobial molecules, several groups aimed to evaluate the in vitro activity of different antimicrobial combinations due to their potentially promising results, especially against multidrug-resistant clinical isolates. In particular, recent studies demonstrated that colistin in combination with different antimicrobial molecules (i.e., rifampicin, meropenem, etc.) exerted potent synergy in vitro against MDR also including colistin-resistant strains [40,130]. However, the discordance of the different in vitro results and the prognostic utility of the synergy tests in the clinical practice still remain controversial [131].

In the last years, the revival of old microbiological techniques such as the serum bactericidal titres (SBTs) has been proposed for monitoring the antibiotic treatment of multidrug-resistant Gram-negative infections [132]. Indeed, SBT is the only laboratory test that integrates drug pharmacodynamics, host pharmacokinetics, and synergistic or antagonistic interactions of antimicrobial combinations into a single index of antimicrobial activity [132]. Recent study demonstrated the high bactericidal activity in serum of patients treated with colistin in association with meropenem rather than other combinations [133]. Although SBT appears to be a promising surrogate PK/PD marker for assessing antimicrobial treatment of Gram-negative infections, the predictive value of the SBT still remains poorly defined, as does the standardization of a more rapid SBT testing method [132].

Novel techniques have been proposed to fight the emergence of antimicrobial resistance, especially among MDR strains. An example is the phenylboronic acid (PBA)-installed micellar nanocarriers, incorporating different antimicrobials that have shown promising in vitro and in vivo results [134]. Recently, Huang and co-workers on a study based on PBA-functionalized micelle loaded with vancomycin and curcumin demonstrated the ability to eradicate drug-resistant bacteria in vitro and in vivo due to the synergism of the two drugs [134]. However, further studies are needed to confirm the results obtained by this novel application.

Lastly, meta-organic framework (MOF)-loaded biohybrid magnetic microrobots have been proposed as an alternative strategy for non-antibiotic bacterial killing [135]. This strategy uses the gradual release of metal ions to cause damage to the bacterial membrane, thus resulting in the efficient killing of bacteria. Although this novel methodology could be considered a promising strategy for treating infections due to MDR strains, further studies are necessary to confirm its clinical utility.

## 6. Conclusions

In the last years, the renewed use of older antimicrobial molecules has revolutionized the treatment of infections due to MDR-GN microorganisms. At the same time, novel approaches, including therapeutic drug monitoring (TDM), for the personalization of the antimicrobial dosage of the different antimicrobial molecules and new therapeutic schemes of treatment, designed by combining antibiotics with limited antimicrobial activity, have revolutionized the treatment of infections due to MDR pathogens.

The clinical usage of colistin alone and in combination with other antimicrobials with scarce and/or limited antimicrobial activity has recently reinvented its role in clinical practice [136,137]. Also, considering the limited antimicrobial options against these pathogens, colistin was defined as the “last-hope resource” for the treatment of DTR infections, especially among critical-ill patients [8]. In this context, the further application of colistin in clinical practice could be utilized in the treatment of emerging pathogens with novel traits of resistance, considering the limited antimicrobial options available.

Conversely, the adverse toxic effects and the limited tissue penetrations in different anatomical districts prompted us to mitigate its role in the clinical setting by limiting its use [1]. In addition, the widespread use of colistin-resistant strains [2,3] poses a serious limitation in terms of the issue of this molecule, especially in light of the new antimicrobial molecules developed recently, possessing high bactericidal activity against MDR microorganisms (i.e., cefiderocol, ceftazidime/avibactam, meropenem/vaborbactam, etc.).

## Figures and Tables

**Figure 1 molecules-29-02969-f001:**
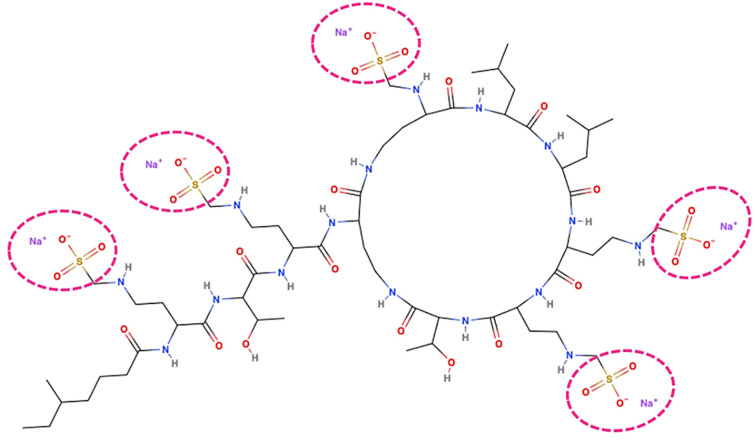
Colistin prodrug structure. Two-dimensional representation of colistin methanesulfonate molecule highlighted the five methanesulfonate groups (inside the purple dotted circles) responsible for the difference between active compound and its colistimethate sodium (prodrug) form. This 2D representation was performed with MolView v2.4 online tool (https://molview.org/ accessed on 1 January 2024).

**Figure 2 molecules-29-02969-f002:**
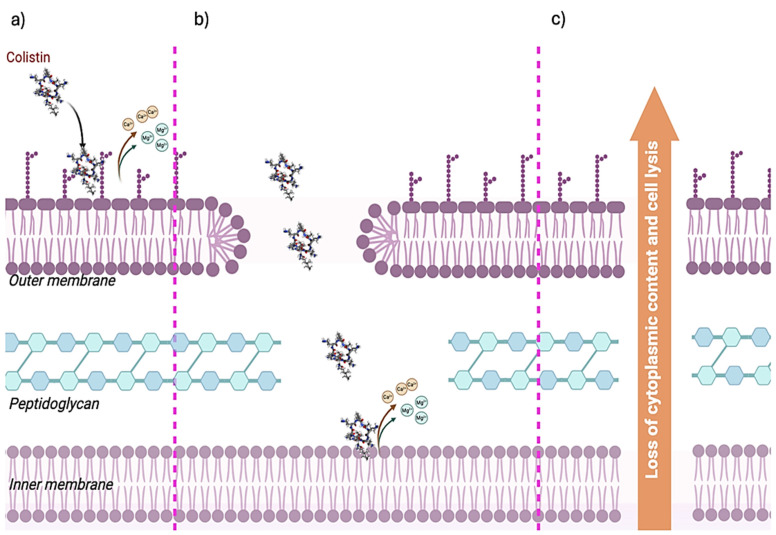
Colistin mechanism of action and SMH model. Schematic representation of colistin’s effect on Gram-negative bacteria. (**a**) Colistin drug acts by competition with Ca^2+^ and Mg^2+^ ions, causing their displacement and leading to (**b**) formation of pores. Colistin goes through the peptidoglycan barrier reaching the inner membrane where it acts by displacing Ca^2+^ and Mg^2+^ ions. All the structural alteration, induced by colistin action on membranes or colistin intracellular accumulation, leads to cell death (**c**). The three-dimensional model of colistin was generated with MolView v2.4 online tool (https://molview.org/ accessed on 1 January 2024). This figure was created in BioRender.com.

**Figure 3 molecules-29-02969-f003:**
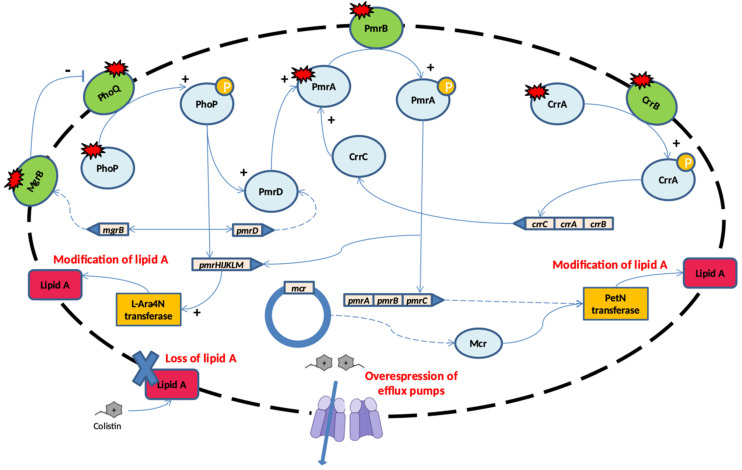
Mechanism of resistance to colistin. Schematic representation of principal colistin resistance mechanisms among Gram-negative bacteria: (i) modification of LPS structure mediated by chromosomal mutation; (ii) modification of LPS structure mediated by plasmid-resistance; (iii) loss of LPS structure; (iv) overexpression of efflux pumps.

**Table 1 molecules-29-02969-t001:** Principal points discussed in this review.

Insight Into the Mechanism of Action	Activity Against Gram-Negative Bacteria	Interaction with Lipid A of LPS	Bacteria Death Induced by Altering Permeability Outer Membrane	Bacteria Death Induced by Oxidative Stress	Bacteria Death Induced by Inhibiting Bacteria Respiration Enzymes
Insight into the mechanism of resistance	Chromosomal resistance				Plasmid resistance
	Reduction in LPS negative charge (i.e., *pmrHIJKLM* operon)	Loss of Lipid A (i.e., Lpx byosinthesis)	Overexpression of efflux pumps (i.e., AcrAB–TolC complex)		Modification of Lipid A structure (i.e., *mcr* gene)
Insight into pharmacokinetic/pharmacodynamic properties	*f*AUC/MIC represent the best PK/PD target for colistin				

**Table 2 molecules-29-02969-t002:** Colistin penetration and assessment of PK/PD target attainment in different sites of infection.

Site of Infection	Study Design	Number of Patients	Setting	Dose	Absolute Tissue Concentrations	Absolute Plasmatic Concentrations	Penetration Rate (AUC_tissue_/AUC_plasma_)	PK/PD Target Attainment	Ref.
Lung	Prospective observational	13	ICU VAP	2 MU q8h IV	Undetectable	C_min_ 1.03 ± 0.69 mg/L AUC/MIC ratio 17.3 ± 9.3 (for MIC = 2 mg/L)	0.00	Suboptimal in ELF	[126]
CSF	Prospective observational	5	ICU	2–3 MU q8h IV	C_min_ 0.47 mg/L AUC 0.53 mg·h/L	C_min_ 9.26 mg/L AUC 10.4 mg·h/L	0.05	Optimal PK/PD target attainment only for *P. aeruginosa* and *A. baumannii* strains exhibiting MIC values up to 0.06 mg/L	[127]
Ocular	Preclinical rabbit uveitis model	20	Uveitis induced after endotoxin injection	5 mg/kg IV	*Aqueous humor* 0.62 ± 0.07 (at 0.5 h) 0.45 ± 0.05 (at 3 h) 0.38 ± 0.08 (at 6 h) *Vitreous humor* 0.02 ± 0.01 (at 3 h)	9.84 ± 2.0 (at 0.5 h) 0.93 ± 0.07 (at 3 h) 0.24 ± 0.08 (at 6 h)	0.07 (aqueous humor at 0.5 h) 0.48 (aqueous humor at 3 h) 1.58 (aqueous humor at 6 h) 0.02 (vitreous humor at 3 h)	Not assessable	[128]

AUC: area under concentration-to-time curve; C_min_: trough concentrations; CSF: cerebrospinal fluid; ELF: epithelial lining fluid; ICU: intensive care unit; IV: intravenous; MIC: minimum inhibitory concentration; PK/PD: pharmacokinetic/pharmacodynamic; VAP: ventilator-associated pneumonia.

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
