# Peer review of "Colistin: Lights and Shadows of an Older Antibiotic"

_molecules, 2024, doi:10.3390/molecules29132969_

Round 1

Reviewer 1 Report

Comments and Suggestions for Authors

In the review paper titled "Colistin: Lights and Shadows of an Older Antibiotic," Diani et al. comprehensively summarize the field literature regarding the history of colistin discovery, its structure and function, mode of action, activity against pathogenic bacterial strains, and its toxicity to humans. The review is well-organized and serves as a valuable resource for researchers in the field as well as the wider scientific community. While I recommend the manuscript for publication, there are several issues and points that need attention prior to its final approval.

The suggestion to authors is to ask english native speaker to edit language prior submitting the manuscript.

Line 28: to to million

Line 72: due to severe its severe adverse effects

Lines 79-80 – I believe hydrophilic and hydrophobic parts are mixed up

Figure 1 shoud be more clear, it appears its in the low resolution and not of the high quality. Is Fig.1. prodrug or colistindrug? These terms should also be more clarified in the text of the manuscript. It would be usefull to readers if functional parts/groups of the colistin structure in figure would be marked and depicted.

Line 114. Again using the word prodrug while it misses clarification and elaboration in the text of the manuscript.

Line 145- in the last year

Line 184- names of the genes shpoud be in italic with small letters (this should be corrected throoughout the text)

Line 186- capital letter marks a protein as a product of a gene (this should be corrected throughout the text)

Line 194 – membrane-bound I believe missing word is a protein

Line 200- A. baumannii lacks all the genes (deliete because)

Line 204 – gene names in italic

Lines 221-222 – clarify

Lines 256-257- elaborate PETN localization

Lines 316-321 – the whole paragraph seams to be repeated

Layout of the paper is landscape beyond page 11.

Line 399-bacterial names shoul be in italic

Line 404 – limited penetration colistin penetration rate…penetration is repeating

Comments on the Quality of English Language

The paper could be improved regarding the language.

Author Response

Reviewer #1

In the review paper titled "Colistin: Lights and Shadows of an Older Antibiotic," Diani et al. comprehensively summarize the field literature regarding the history of colistin discovery, its structure and function, mode of action, activity against pathogenic bacterial strains, and its toxicity to humans. The review is well-organized and serves as a valuable resource for researchers in the field as well as the wider scientific community. While I recommend the manuscript for publication, there are several issues and points that need attention prior to its final approval.

The suggestion to authors is to ask english native speaker to edit language prior submitting the manuscript.

Authors’ reply and amendments: We thank the reviewer for his comments. The manuscript was revised by a native english speaker as requested

Line 28: to to million

Authors’ reply and amendments: The sentence was modified as requested

Line 72: due to severe its severe adverse effects

Authors’ reply and amendments: The sentence was modified as requested

Lines 79-80 – I believe hydrophilic and hydrophobic parts are mixed up

Authors’ reply and amendments: The sentence was modified as requested

Figure 1 shoud be more clear, it appears its in the low resolution and not of the high quality. Is Fig.1. prodrug or colistindrug? These terms should also be more clarified in the text of the manuscript. It would be usefull to readers if functional parts/groups of the colistin structure in figure would be marked and depicted.

Authors’ reply and amendments: The caption was modified by adding the correct term and a high resolution file of the figure is uploaded separately. Also, the methanesulfonate groups are better shown in the figure as requested

Line 114. Again using the word prodrug while it misses clarification and elaboration in the text of the manuscript.

Authors’ reply and amendments: The prodrug form was explained in the text as requested

Line 145- in the last year

Authors’ reply and amendments: The sentence was modified as requested

Line 184- names of the genes shpoud be in italic with small letters (this should be corrected throoughout the text)

Authors’ reply and amendments: the genes name was modified as requested

Line 186- capital letter marks a protein as a product of a gene (this should be corrected throughout the text)

Authors’ reply and amendments: The protein and operon names were checked and modified as requested

Line 194 – membrane-bound I believe missing word is a protein

Authors’ reply and amendments: The sentence was modified as requested

Line 200- A. baumannii lacks all the genes (deliete because)

Authors’ reply and amendments: the list of the genes are cited in the previous sentence and in A.baumannii. We opted to retain the sentence in this format for the readers

Line 204 – gene names in italic

Authors’ reply and amendments: the gene and operon names were modified as requested

Lines 221-222 – clarify

Authors’ reply and amendments: The sentence was modified as requested

Lines 256-257- elaborate PETN localization

Authors’ reply and amendments: The localization of mcr gene was added to the text as requested

Lines 316-321 – the whole paragraph seams to be repeated

Authors’ reply and amendments: The sentences referred to two different bacteria infections (p. aeruginosa and A.baumanii) as discussed in the text

Layout of the paper is landscape beyond page 11.

Authors’ reply and amendments: The layout of the paper was modified as requested

Line 399-bacterial names shoul be in italic

Authors’ reply and amendments: The bacterial names were modified as requested

Line 404 – limited penetration colistin penetration rate…penetration is repeating

Authors’ reply and amendments: The sentence was modified as reuqested

Reviewer 2 Report

Comments and Suggestions for Authors

Dear Authors,

This review delves into the fundamental principles governing the mode of action of colistin, highlighting its pertinent resistance characteristics and clinical applications from both pharmacological and practical standpoints. The authors have comprehensively outlined the mechanism of action, the emergence of resistance determinants, and the optimal strategies for administering colistin in the management of challenging infections caused by multidrug-resistant (MDR) Gram-negative bacteria. In the concluding section, given the paucity of effective antimicrobial agents against these pathogenic organisms, colistin is designated as the "last resort option" for treating difficult-to-resolve (DTR) infections, particularly among critically ill patients. Based on these considerations, I recommend that the review be published after addressing the following corrections:

1. In the keywords: .Colistin should be added.

2. The spaces between numbers and units in some parts of the article need to be added.

3. The formats of the reference are incorrect, such as lack page numbers, such as 3, 4, 5,.....

Author Response

Reviewer #2

Dear Authors,

This review delves into the fundamental principles governing the mode of action of colistin, highlighting its pertinent resistance characteristics and clinical applications from both pharmacological and practical standpoints. The authors have comprehensively outlined the mechanism of action, the emergence of resistance determinants, and the optimal strategies for administering colistin in the management of challenging infections caused by multidrug-resistant (MDR) Gram-negative bacteria. In the concluding section, given the paucity of effective antimicrobial agents against these pathogenic organisms, colistin is designated as the "last resort option" for treating difficult-to-resolve (DTR) infections, particularly among critically ill patients. Based on these considerations, I recommend that the review be published after addressing the following corrections:

1. In the keywords: .Colistin should be added.

Authors’ reply and amendments: We thank the reviewer for his comments. The term was added to the keywords as requested

2. The spaces between numbers and units in some parts of the article need to be added.

Authors’ reply and amendments: The manuscript was completely revised to remove or add spaces between number as requested

3. The formats of the reference are incorrect, such as lack page numbers, such as 3, 4, 5,..…

Authors’ reply and amendments: The sentences were modified as requested

Reviewer 3 Report

Comments and Suggestions for Authors

In this manuscript, the authors were focused on colistin, and summarized the action mode, resistance determinants and optimal administration in infection treatment. The topic seems to be interesting and this work can be useful for this field. However, the following problems should be addressed before further consideration of publication:

1. The keywords should be revised, and “colistin” can be added.

2. In the Introduction section, a scheme can be created to better summarize the detailed mechanism and application property of colistin.

3. In the Introduction section, typical examples of bacterial killing can be briefly introduced. The recent advances in this fields should be contained including: 10.1016/j.colsurfa.2024.133295, 10.1002/EXP.20210145.

4. As a comprehensive review, composite figures of typical researches in the related fields or schematic illustrations should be added. At least 3-5 figures should be included for better demonstration. The current manuscript is not rich enough in graphic materials and convincing descriptions.

5. The authors are suggested to add a separate section to describe challenges and future perspectives for researches and applications of colistin.

6. In the manuscript, the depth could be improved if the authors provided some insights of micro-/nano and biological interactions responsible for colistin resistance.

7. The references should be updated considering some novel researches in recent 3-5 years.

Author Response

Reviewer #3

In this manuscript, the authors were focused on colistin, and summarized the action mode, resistance determinants and optimal administration in infection treatment. The topic seems to be interesting and this work can be useful for this field. However, the following problems should be addressed before further consideration of publication:

1. The keywords should be revised, and “colistin” can be added.

Authors’ reply and amendments: The term was added to the keywords, as requested

2. In the Introduction section, a scheme can be created to better summarize the detailed mechanism and application property of colistin.

Authors’ reply and amendments: We thank the reviewer for his comments. We add to the introduction section a summary of the principal points discussed in this review with their principal aspects, as requested

3. In the Introduction section, typical examples of bacterial killing can be briefly introduced. The recent advances in this fields should be contained including: 10.1016/j.colsurfa.2024.133295, 10.1002/EXP.20210145.

Authors’ reply and amendments: The references were added to the text as requested

4. As a comprehensive review, composite figures of typical researches in the related fields or schematic illustrations should be added. At least 3-5 figures should be included for better demonstration. The current manuscript is not rich enough in graphic materials and convincing descriptions.

Authors’ reply and amendments: Novel figures and Table were added to the manuscript as requested

5. The authors are suggested to add a separate section to describe challenges and future perspectives for researches and applications of colistin.

Authors’ reply and amendments: We thank the reviewer for his comments. As requested, we discussed further directions in the discussion section. However, we opted to maintain the review in this format by not adding a separate paragraph for “future perspectives for researches and applications of colistin” for clarity of the review and to facilitate the readers

6. In the manuscript, the depth could be improved if the authors provided some insights of micro-/nano and biological interactions responsible for colistin resistance.

Authors’ reply and amendments: We thank the reviewer for his comments. However, we retain to not add sentences regarding the interaction of micro and nano to this review because is out of the focus of this review. Thus, we opted to maintain the review in this format

7. The references should be updated considering some novel researches in recent 3-5 years.

Authors’ reply and amendments: Recent studies published in the last two years were added the discussion section as requested

Round 2

Reviewer 3 Report

Comments and Suggestions for Authors

I have checked the manuscript yet the following problems are not addressed considering the comments in last revision:

1. In the Introduction section, typical examples of bacterial killing can be briefly introduced. The recent advances in this fields should be contained including: 10.1016/j.colsurfa.2024.133295, 10.1002/EXP.20210145.

2. The description depth of challenges and perspectives could be further improved with detailed explanation in this section.

3. The manuscript should be reviewed thoroughly considering some spelling and format errors.

4. The references should be updated considering some novel researches in recent 3-5 years.

Author Response

I have checked the manuscript yet the following problems are not addressed considering the comments in last revision:

1. In the Introduction section, typical examples of bacterial killing can be briefly introduced. The recent advances in this fields should be contained including: 10.1016/j.colsurfa.2024.133295, 10.1002/EXP.20210145.

Authors’ reply and amendments: We thank the reviewer for his comments. The suggested references were added to the text as requested. In particular, we added the suggested references to a novel section named “5. Future perspectives” as suggested by the reviewer in the previous reviewing.

2. The description depth of challenges and perspectives could be further improved with detailed explanation in this section.

Authors’ reply and amendments: A novel section named “5. Future perspectives” was added to the manuscript to better discuss this important point raised by the reviewer. In this section we discuss novel methodologies, hypothesis of novel studies and detailed explanation of different techniques, as requested

3. The manuscript should be reviewed thoroughly considering some spelling and format errors.

Authors’ reply and amendments: The manuscript was completely checked and revised for misspelling and grammatical errors

4. The references should be updated considering some novel researches in rec

Authors’ reply and amendments: Novel studies were added to the references as requested